# Participation in Organized Sports and Self-Organized Physical Activity: Associations with Developmental Factors

**DOI:** 10.3390/ijerph16040585

**Published:** 2019-02-18

**Authors:** Nora Wiium, Reidar Säfvenbom

**Affiliations:** 1Department of Psychosocial Science, Faculty of Psychology, University of Bergen, Christies gate 12, 5020 Bergen, Norway; 2Department of Physical Education, Norwegian School of Sport Sciences, Sognsveien 220, 0806 Oslo, Norway; reidar.safvenbom@nih.no

**Keywords:** organized sports, self-organised physical activity, demographics, developmental factors, Norway

## Abstract

Engagement in organized sports is associated with developmental factors, such as, healthy growth, cognitive abilities, psychological well-being and lower substance use. Research also suggest that the spontaneous free play that characterises self-organized physical activity (PA) provides young people with opportunities to learn social skills, such as self-regulation and conflict-resolution skills. We assessed associations between participation in the two activity types and several demographics along with developmental factors (e.g., body mass index (BMI)). Data was from a representative sample of 2060 students attending 38 schools in Norway (mean age (*M_age_*) = 15.29, standard deviation (*SD*) = 1.51; 52% females). Results indicated that while engagement in organized sports was more related to developmental factors, relative to self-organized PA, engaging concurrently in both activities for at least an hour a week was more developmentally beneficial than engaging only in one for the same amount of time. Thus, PA programmes for students will enhance their effectiveness if they focus on structured activities but also self-organized activities where students can coordinate themselves.

## 1. Introduction

Physical activity (PA) has received much attention in research, with a primary focus on the health benefits of the behaviour. Immediate and long-term health benefits that have been observed include a sense of well-being, healthy muscles, bones and joints, reduced risk of developing diabetes, cardiovascular diseases and some cancers, and a longer healthy life [1,2,3]. Among young people in particular, PA has been associated with several positive outcomes including physical [4,5], psychological [5,6,7] and social outcomes [8,9]. In addition, an extensive literature review undertaken by Morris, Sallybanks, and Willis [10] revealed that well-structured sports and physical activity programmes, which operate in safe and engaging environment and which engage the involvement and leadership of young people, protect against antisocial behaviours. 

Despite its numerous benefits, research shows that physical activity declines during adolescence [11,12,13] and that programmes established to increase the activity level among adolescents are challenging to conduct [14]. Compared to childhood, adolescence is more complex when it comes to behaviour and development, and determinants as well as effects of physical activity are difficult to identify. There is a great deal of uncertainty regarding the relationship between demographic factors (e.g., ethnicity, social class, parenthood and gender), type of activity (organized sports vs. self-organized physical activity), amount of this involvement, and developmental health benefits. In the present study, we assess several associations between major demographic variables, activity types and a sample of developmental factors. 

### 1.1. Involvement in Physical Activity (PA): The Role of Demographic Factors

Gender, ethnicity, social class and parental support for PA have been found to play a relevant role in youth’s participation in PA in earlier research. The issue of gender has been debated in sports science for decades and research shows that men have dominated traditional sports at different levels while women have been fighting for access among facilitators as well as in the field of practice [15]. Also, modern sports, often defined as lifestyle sports [16,17] are most often performed by men [18,19], although women have found their own paths into some of these sports, for example, snowboarding [20] and windsurfing [21]. During adolescence, boys are more physically active than girls [12] and studies from Norway indicate that girls are less involved in both organized sports [22] and physical activity in general [13]. 

Ethnic minority groups and females from some of these groups in particular, are less involved in many types of physical activity contexts than ethnic majority groups [23,24]. In Norway, not more than 9% of Norwegian-Pakistani females reported membership in a local sports club [25]. However, more recent reports indicate that minority groups do not report significantly less activity outside organized contexts; besides, minority males may even spend more time in fitness centres than their counterparts who are involved in organized sports [22]. 

Findings on the importance of social class for youth participation in sports are mixed. For some sports, social class is still a palpable predictor, while for others, it is not; nevertheless, Green [26] suggest that there is still evidence to predict a general correlation between social class and youth sports participation. Parents’ social, cultural and economic capital appear to predict parents’ social support for involvement in PA and sports [27,28], and the role of social support from parents has gained more attention as a major predictor for involvement in PA [28,29]. Activity support from parents may overrule most other predictors for children and adolescents’ involvement in both organized sports and PA in general. In line with this, Stroot [30] concluded that “if the parents are not actively involved, nor intentionally provide sporting experiences for the child, the chance that the child will be exposed to the sporting world at an early age is limited” (p. 131). To ascertain the relevance of demographic factors in sports and PA participation in the Norwegian context, we assessed the influence of gender, age and several parental factors, such as parental income and activity support.

### 1.2. Physical Activity Types and Developmental Factors

Health benefits can be attained only with a sufficient amount of PA and the World Health Organisation (WHO) recommends at least an hour of moderate to vigorous physical activity per day for children and adolescents [31]. The recommendations are based on studies derived from the “Dose–Response curve” [32], which explains the relationship between dose of physical activity (in terms of MET: metabolic equivalent of task) and the biological response to this dose in terms of prevention of diseases, such as diabetes and coronary heart diseases. 

In the physical activity and youth literature, a distinction has often been made between organized sports and unorganized (or self-organized) physical activity (PA) [33]. The former usually refers to types of PA organized in educational institutions, sports clubs or other organizations that are coordinated by adults; thus, organized sports have some form of structure around the activities that young people engaged in. Alternatively, unorganized PA, which is sometimes referred to as free play, is less structured and is usually coordinated by young people themselves. However, Säfvenbom, Wheaton, and Agans [34] argued that the dominant understanding of lifestyle sport as unorganized and unstructured is a misinterpretation. Based on their observations of four lifestyle sport contexts, the authors labelled these as self-organized, self-determined, self-structured, collaboratively supervised, and relationally process-oriented. Despite the fact that PA outside organized sports varies in terms of how it is performed and organized, we use the term “self-organized” as an alternative to unorganized PA from here.

Organized youth sport is one of the most common leisure-time activities during adolescence in Norway [35]. Activities are provided in local sports clubs, which are organized within the national non-governmental sports association (the Norwegian Olympic and Paralympic Committee and Confederation of Sports (NIF)). Even if commercial activity arenas have become more popular during the last two decades, around 50% of all 15 year olds are still members of a public sports club [22]. 

Studies show that self-organized PA and lifestyle sport contexts, in particular do affect adolescents differently [34,36]. Generally, participation in organized leisure-time activities is associated with positive developmental outcomes, such as better physical performance, academic achievement, educational attainment, psychological adjustment, connection with positive peer network, enhanced youth identity as well as lower rates of antisocial behaviours [7,37]. These positive outcomes appear to be even more pronounced for youth from low-income families who participate in organized activities [37]. In contrast, unsupervised leisure time has sometimes been linked to negative youth development [38] and it appears to be a conventional belief that “adolescent participation in structured activities, meaning those with adult leaders, regular meetings, and skill-building activities, is related to good adjustment” while “participation in unstructured, unsupervised, peer-oriented activities is related to poor adjustment” [39] (p.197). 

Despite the numerous advantages associated with organized leisure-time activities, it has been argued that the participation fee may create a social selection situation where only young people from affluent families have access to the activities [40]. Moreover, the competition, discipline and hard work involved in several of these organized activities during early adolescence may undermine young people’s motivation and thus active participation [41]. As argued by Støckel, Strandbu, Solenes, Jørgensen, and Fransson [42], there is an inherent strain in organized sports, which is particularly present for children and youth. This may undermine the positive outcomes derived from organized activities. Moreover, organized youth sport is seen as an arena, which promotes social integration and offers physical benefits from one’s participation, yet the inherent logic of competition in sports potentially creates losers and dissatisfaction and the notion that organized sport is a developmental asset for all could be a great myth [41,43]. 

Despite research claiming that self-organized sport contexts can be questioned regarding the opportunity to increase involvement from hard to reach groups [44], self-organized activity in general can be less expensive or without a participation fee and may thus be undertaken anywhere by many young people. The positive outcomes that are associated with self-organized activity are thus not available to only a few privilege ones but can be experienced by those who do not feel at home in organized and competitive settings, such as organized youth sports. Earlier studies also suggest that the spontaneous free play that characterises self-organized PA, in particular lifestyle sports, may provide young people with opportunities to learn social skills, such as independence, self-regulation skills, cooperation and problem-solving skills [34,45,46]. According to Säfvenbom and colleagues [34], such “contexts may produce thriving young people, not in spite of, but because of involvement in contexts with no strict rules, no formal leaders and no a priori competence or performance goals” (p. 16).

Furthermore, the fun factor and imaginative play, common in many self-organized PA are more likely to promote creativity, identity and self-actualization [47]. Because of factors, such as the fun factor and an inherent desire to play in children and young people, self-organized PA may be more intrinsically motivated. Thus, it may be seen as important for the experience of personally meaningful activity, enjoyment, autonomy, self-determination, development of competence and self-validation [48]. In summary, engagement in organized sports and self-organized PA has been associated with several developmental factors although the benefits appear to vary according to the type of activity.

### 1.3. The Present Study

The literature review suggests that both organized sports and self-organized PA are associated with several developmental factors. However, not much has been done regarding research on developmental benefits when both activity types are concurrently engaged in, a step we address in the present study. Specifically, we explore (1) the association between demographic variables (i.e., gender, ethnicity, parents’ income (as an indicator of social class), and parental activity support) and participation in organized youth sports, self-organized PA, as well as participation in both forms of activity; (2) how organized sports and self-organized PA among young people, either in combination or alone are associated with developmental factors, such as physical development (body mass index (BMI)), academic achievement, perceived academic competence, perceived social competence, smoking and alcohol use. Based on the above review of the general benefits of physical activity [7,34,37,45,46], we hypothesize that optimal scores in developmental factors will be associated with a combination of organized sports and self-organized PA more than each one of them alone.

## 2. Methods

The present study builds on data collected from a cross-sectional study that was conducted as part of the “Goodness of fit in Norwegian Youth Sport” study in 2008. Due to research showing that sport participation (not considering dropout from organized sports) has been stable the last three decades, we consider the data from this set as valid.

### 2.1. Sample and Data Collection

A representative sample of 2971 students from 38 different schools in Norway were stratified according to school-level and geographical area. A total of 2060 (71%) adolescents (995 males and 1060 females; 5 did not indicate their sex) from ages 13 to 18 (mean age (*M_age_*) = 15.29, standard deviation (*SD*) = 1.51) completed the self-report questionnaire during school time. The NSD—Norwegian Centre for Research Data approved the study, while parents and students gave their written informed consent before students could participate in the study. 

### 2.2. Measures

*Organized sports and self-organized PA.* The items used in measuring organized sports and self-organized PA were an adaptation of Sagatun, Søgård, Selmer, Bjertness, and Heyerdahl [49] measurement of PA. For organized sports, participants were asked: “How many hours per week do you train or compete enough to make you sweat or breathe hard?” and for self-organized PA, participants were asked: “How many hours per week do you play or exercise enough to make you sweat or breathe hard?” Responses to both questions were as follows: (1) 0, (2) 1–2, (3) 3–4, (4) 5–7, (5) 8–10 and (6) 11 h or more per week. When the two activity types were treated as binary variables in data analysis, 0 h of activity was recoded as (0) non-participation, while one or more hours of PA were recoded as (1) participation. In addition, the two activity types were combined to create two composite variables: (a) participation in both organized sports and self-organized PA versus participation in only organized sports; and (b) participation in both organized sports and self-organized PA versus participation in only self-organized PA.

*Parents’ total income.* Participants were asked to indicate what they thought were their parents’ total income (in Norwegian currency) per year. The options were: (1) Less than 200,000; (2) 200,000–400,000; (3) 400,000–800,000; (4) 800,000–1 million; and (5) More than 1 million.

*Mother and father’s ethnic background.* Participants were asked to indicate whether their (a) mother and (b) father were from (1) Norway; (2) other country in Europe; (3) America; (4) Asia; (5) Africa; (6) Australia and (7) don’t know. For the analysis, the responses were recoded into (1) Norwegian, (2) not Norwegian, and missing when participants indicated that they do not know. 

*Parental activity support.* Verbal and behavioural support from parents regarding physical activity and exercise in everyday life, weekends and holidays were assessed with five items [41]: (a) How often do your parents encourage you to play, exercise or engage in sports? (b) How often are your parents with you when competing in sports or engaging in physical activity? (c) How often do you engage in sports or play together with your mother or father? (d) How often do your parents take you out for outdoor activities during weekends? (e) How often do your parents take you out for outdoor activities during holidays? Responses were measured on a 4-point Likert scale, with responses ranging from (1) Almost never or never, to (4) Every day. Cronbach’s alpha of the items was 0.70.

*Body mass index (BMI).* Participants’ height and weight measures that were used to calculate BMI were self-reported. BMI was calculated using Cole and colleagues’ procedure where they adjust for age and gender [50].

*Perception of academic competence.* The Academic Competence subscale from the Norwegian revised version [51] of Harter’s Self-Perception Profile for Adolescents (SPPA; [52]) was used to assess participants’ perceptions of their academic competence. Participants indicated the extent to which they identified with the following 5 items about their academic activities or ability: (a) I think I am as smart as others my age; (b) I am pretty slow to complete my school work; (c) I do very well at school; (d) I have difficulty answering questions correctly at school; and (e) I consider myself pretty intelligent. Responses to these items were (1) Describes me very poorly, (2) Describes me fairly poorly, (3) Describes me fairly well, and (4) Describes me very well. For the analysis, all items were framed in the positive direction, such that higher scores meant higher academic perceptions. Cronbach’s alpha of the items was 0.72.

*Academic grades.* Participants were asked to indicate their current academic grades (1 to 6) on four school subjects: mathematics, English, physical education and Norwegian.

*Perception of social competence.* The Social Competence subscale from the Norwegian revised version [51] of the (SPPA; [52]) was used to assess participants’ perceptions of their social competence. The 5 items that make up the scale are as follows: (a) I find it quite difficult making friends; (b) I have many friends; (c) Other young people have a hard time liking me; (d) I am popular with peers; and (e) I feel that I am accepted by my peers. Participants indicated whether the statements: (1) Describes me very poorly, (2) Describes me fairly poorly, (3) Describes me fairly well, and (4) Describes me very well. All items were framed in the same direction, such that higher scores indicated higher levels of perceived social competence. Cronbach’s alpha of the items was 0.78.

*Participants’ smoking and alcohol status.* To assess the status of smoking, participants were asked to indicate whether (1) they have never smoked before; (2) they have tried to smoke on one or few occasions; (3) they smoke sometimes, but less than 10 times a month; (4) they smoke almost every day or on a regular basis. For alcohol use, participants indicated whether (1) they have never used alcohol; (2) they have used alcohol but have never been drunk; and (3) they drink alcohol to get drunk.

### 2.3. Statistical Analysis

Frequency distribution was conducted on each of the two activity variables as well as on the two composite activity variables that were created. Simple descriptive statistics (i.e., mean, standard deviation and range) and correlations were estimated for all study variables. The three variables that were measured with 5 items each (i.e., parental activity support, perception of academic competence and perception of social competence) were included in the data analysis as mean scores. Here, participants who had at least 80% of the responses to the five items that were used to assess each of the three variables were included in the computation of the mean scores. Descriptive analyses were carried out with IBM SPSS Statistics for Windows, version 24 (IBM Corp., Armonk, NY, USA). Four regression analyses were conducted in Mplus statistical programme, version 7 [53] to assess how the demographic and developmental factors were associated with the activity variables:
Ordinal regression analysis with organized sports as an ordered categorical variable;Ordinal regression analysis with self-organized PA as an ordered categorical variable;Logistic regression analysis with participation in both activity types for at least one hour versus participation in only organized sports for the same time as the outcome variable;Logistic regression analysis with participation in both activity types for at least one hour versus participation in only self-organized PA for the same time as the outcome variable.


Montecarlo integration was used in the Mplus analyses due to the ordered categorical outcome variables and missing cases in the study variables [53]. The majority (91%) of the participants had missing data on six or less study items. The full information maximum likelihood (FIML) estimator with robust standard errors (MLR; Robust Maximum Likelihood) was used in all regression analyses to handle missing data. FIML uses all available data and assumes that missing data is at least missing at random (MAR). This implies that the probability of a missing data of a certain variable Y relies on other measured variables in the data, but not the values of variable Y itself. This is assumed to be the case in our data as the *p*-value of the Little’s MCAR (Missing Completely at Random) test we conducted was less than 0.05.

## 3. Results

### 3.1. Descriptive Analysis

A frequency distribution of the physical activity variables revealed that 32.4% of the participants did not engage in organized sports. About 31% engaged in the activity for 1–4 h per week, while 36.4% engaged in the activity for at least 5 h per week. For self-organized PA, 22% did not engage in the activity, 53% engaged in the activity for 1–4 h per week, while 25.5% engaged in the activity for at least 5 h per week. The proportion of participants who engaged in both activities for at least 1 h per week was 57.5% while 11.3% did not engage in any of the activity types (Table 1).

### 3.2. Correlation Analysis

In correlation analysis, gender correlated significantly with organized sports and self-organized PA, *r* = −0.06, *p* < 0.01 and *r* = −0.12, *p* < 0.01, respectively (both in males’ favour), but not with participation in both activities versus participation in only one activity (Table 2). Furthermore, while participation in organized sports decreased with age (*r* = −0.17, *p* < 0.01), there was no significant correlation between age and participation in self-organized PA. Positive correlations were found between parents’ total income, parental activity support and the activity variables (Table 2). Furthermore, participants with a Norwegian mother or father were significantly more likely to engage in organized sports. For self-organized PA, the correlation was significant, albeit small when participants reported that they had a Norwegian mother (*r* = −0.06, *p* < 0.05). BMI scores correlated negatively with participation in organized sports (*r* = −0.10. *p* < 0.01), while no such significant correlation was found for self-organized PA.

Positive correlations were found between organized sports and academic grades in all school subjects, except for English (Table 2). In addition, small but positive correlations were found between self-organized PA and three of the academic variables: academic competence, grade in mathematics, and grade in physical education (*r* = 0.09. *p* < 0.01, *r* = 0.06, *p* < 0.05 and *r* = 0.17, *p* < 0.01, respectively). Positive correlations were observed between physical education grade and the composite activity variables (i.e., both activity types versus only one activity) while negative correlation was found between grade in English and one of the composite activity variables (i.e., both activity types versus only self-organized PA). Significant correlations were found between social competence and all the activity variables. In addition, there were negative correlations between smoking status and engagement in the activity types. Similar results were found for alcohol use except for the correlation between the variable and self-organized PA, which was not statistically significant (Table 2).

### 3.3. Ordinal Regression Analyses of Organized Sports and Self-Organized PA 

In Table 3, findings from ordinal regression analyses of organized sports and self-organized PA as two separate outcome variables are presented. For organized sports, the findings suggest that among demographic variables, only age and parents’ total income remained significantly associated with the activity when all the study variables were examined simultaneously. Thus, younger participants as well as participants who reported higher parental income engaged in organized sports more often than older participants and those who reported lower levels of parents’ income (*β* = −0.21, *p* < 0.001 and *β* = 0.07, *p* < 0.01, respectively). For the developmental factors, BMI, physical education grade and smoking status remained significant; thus, participants with “healthy” weight, those with higher grades in physical education and those who smoked less often were more likely to engage in organized sports (*β* = −0.05, *p* < 0.05, *β* = 0.38, *p* < 0.001 and *β* = −0.12, *p* < 0.001, respectively). In correlation analysis, there was no significant relationship between English grade and organized sports, but in regression analysis, a negative but significant association was found, albeit small. Furthermore, a negative correlation was observed between alcohol use and organized sports. However, in ordinal regression, positive association was observed, perhaps indicating a suppression effect (Table 3). 

For self-organized PA, the demographic variables: gender, age and parents’ total income were significantly related to the activity (*β* = −0.08, *p* < 0.01, *β* = −0.10, *p* < 0.01 and *β* = 0.08, *p* < 0.01, respectively), together with physical education grade and smoking status (*β* = 0.15, *p* < 0.001 and *β* = −0.08, *p* < 0.05, respectively). Like organized sports, suppression effect was also found for the influence of alcohol use on self-organized PA (Table 3). 

### 3.4. Logistic Regression Analyses of Participation in Both Activities versus Only One Activity

In logistic regression (Table 4), associations between developmental factors and participation in both activity types versus participation in only organized sports were examined. Age, parents’ total income and physical education grade were the only study variables that remained significant. Thus, younger participants, participants who reported higher income of parents and those who reported higher grades in physical education were more likely to report that they participated in both organized sports and self-organized PA than in only organized sports (odds ratio (OR) = 0.87, 95% confidence interval (CI) 0.79–0.96; OR = 1.20, 95% CI 1.05–1.38; and OR = 1.47, 95% CI 1.22–1.77, respectively). When the outcome was participation in both activities vs. participation in only self-organized PA, age, grades in English and physical education as well as smoking status were significant. Here, younger participants, participants who reported lower grades in English, those who reported higher grades in physical education, and those who were less likely to smoke were more likely to engage in both activities than in only self-organized PA (OR = 0.69, 95% CI 0.63–0.75; OR = 0.81, 95% CI 0.70–0.94; OR = 2.71, 95% CI 2.24–3.27; and OR = 0.74, 95% CI 0.63–0.87, respectively) (Table 4).

## 4. Discussion

### 4.1. General Findings

Our aim was to explore the association between various demographic variables and participation in organized youth sports, self-organized PA, as well as participation in both forms of activity. We also examined how organized sports and self-organized PA, either in combination or alone were associated with several developmental factors. The analyses of data from a representative sample of Norwegian adolescents (age 13–18) showed that the majority of participants engaged in organized sports although about a third did not. Similarly, the majority participated in self-organized PA whilst almost a quarter did not. Nearly three fifths engaged in both activity types while about a tenth were not involved in any physical activity in their leisure time. Correlation analysis showed that most of the study variables (i.e., demographics and developmental factors) were related to organized sports. However, when testing the relations, only age, parents’ total income, BMI, physical education grade and smoking status remained significantly associated with organized sports. Similarly, for self-organized PA, only sex, age, parents’ total income, physical education grade and smoking status remained significantly associated with the activity. 

When the dependent variable was participation in only organized sports versus participation in both activities, age, parents’ total income and physical education grade were the only variables that maintained their significant associations in regression analysis. Thus, younger participants, those who reported higher parental income and those who had higher grades in physical education were more likely to participate in both organized sports and self-organized PA for at least an hour compared to participation in only organized sports for the same time. Furthermore, when the dependent variable was participation in only self-organized PA versus participation in both activity types, only age, English grade, physical education grade and smoking status retained their significant associations. In particular, younger participants, those who reported lower grades in English, those who reported higher grades in physical education, and those who were less likely to smoke participated in both activity types for at least one hour compared to participation in only self-organized PA for the same time. Judging from the overall results, our hypothesis that a combination of organized sports and self-organized PA will be associated with more developmental factors than each one of the activities alone was confirmed to some extent.

### 4.2. Demographics, Developmental Factors and Activity Types

While the demographics: sex, age, parents’ ethnicity, parents’ income and parental activity support were generally associated with participation in organized sports and self-organized PA in correlation analysis, only age, sex and parents’ income were associated with the activity types to varying degrees in regression analysis. Previous findings on gender differences in PA have been mixed, although several studies suggest that males tend to be involved in PA more than females [12,54,55]. Several reasons have been given for the gender difference. For example, Chalabaev, Sarrazin, Fontayne, Boiché, and Clément-Guillotin [56] argued that the gender difference favouring males was not only an internalization of stereotypes and gender roles during the socialization process with significant others, but also a situational influence where the presence of stereotypes in the environment affect cognition, motivation and behaviour. The present observation in our Norwegian sample that males engaged in self-organized PA more than females while no gender difference was found for organized sports (i.e., structured and supervised) is somehow consistent with Chalabaev and colleagues’ [56] assertion. Moreover, previous findings suggest that the general activity level of girls is more influenced by organized sports activities compared to boys [54]. 

The current findings that the activity types (i.e., both organized sports and self-organized PA) tend to decrease with age among young people has also been found in previous studies [11]. From the current study, this decrease could be due to less parents’ activity support, which in line with results from correlation analysis was more likely to be reported by older participants. Indeed, it is logical to argue that parents are more likely to engage in PA with their children or take them out for outdoor activities the younger children are. Nevertheless, as parents’ activity support was related to participation in organized sports and self-organized PA although only in correlation analysis, older adolescents may benefit from this parental resource as well. Furthermore, while having a Norwegian mother or father was associated with both activity types, this finding was not conclusive, as it was only observed in correlation analysis. However, it is possible that the influences of parents’ ethnic status as well as that of parents’ activity support on the activity types were mediated by age, sex and parents’ income, which appeared to be the strongest demographic factors, as they remained significant in regression analysis. This mediation can be worthwhile assessing in future studies to determine the significance of parental factors on adolescents’ engagement in sports and PA. 

For the developmental factors, although several of them were associated with the activity types in correlation analysis, only BMI and physical education grade were statistically significant in regression analyses. However, BMI was only related to organized sports. Previous findings on the relationship between BMI and PA for young people has been equivocal [57,58]. In a longitudinal study, Cairney and Veldhuizen [57] even observed a bidirectional relationship between BMI and organized sport participation, although the two were weakly related. While BMI was also weakly related to organized sports in our cross-sectional study, the current findings in line with previous ones suggest that organized sports activities may somewhat contribute to healthy BMI. Moreover, physical education grade was related to both organized sports and self-organized PA. Higher grades in physical education may indicate athletic competence, enjoyment as well as social support from peers and adults, indicators that turn to reflect intrinsic motivation [59]. Intrinsic motivation in physical education may thus translate into increased participation in both structured sports and unstructured PA as was observed in the present study. It is probably not surprising that physical education grade was associated with engagement in both organized sports and self-organized PA. However, earlier findings of the other developmental factors (e.g., social competence) were not definite in our Norwegian sample, especially when examined together. More research is therefore needed to shed light on the specific relations of these factors with physical activity.

For smoking and alcohol use, negative associations with the activity types were observed although this finding was more consistent for smoking. For alcohol use, a suppression effect was found in the regression analysis as the direction of its associations with the activity types changed from negative to positive. Moreover, previous studies have mostly found negative associations with leisure time organized activities for smoking and not alcohol use [60,61,62]. Audrain-McGovern, Rodriguez, Cuevas, and Sass [63] argued that the reward gained from physical activity functions as a process where engaging in the behaviour reduces the initiation and progression of smoking. Thus, it appears that engaging in physical activity protects against smoking more than it does against alcohol use. This interesting finding can be explored further in future studies.

### 4.3. Participation in Both Activities versus Participation in Only One Activity 

Our hypothesis that participation in both organized sports and self-organized PA compared to participation in only one activity will be associated with more developmental factors was confirmed for grade in physical education and smoking status. Students who reported higher grades in physical education were also more likely to engage in the two activity types rather than in only one. The earlier argument that athletic competence and thus intrinsic motivation in physical education may be related to increased participation in organized sports and self-organized PA may apply here as well. Rather interesting is the finding that lower grades in English were associated with participation in both activity types compared to participation in only self-organized PA (in correlation analysis). Besides, when the activity types were analysed separately in regression analysis, English grade was also negatively associated with participation in organized sports. It is not clear whether this has something to do with the language of the majority or minority. Norwegian is the language of the majority although many young people in Norway are also fluent in English. The use of majority or minority language could be an important topic to consider in research on physical activity among young people. 

For smoking, when students engaged in both activity types rather than in only self-organized PA, they were less likely to smoke. This observation was not made when participation in both activity types was compared to participation in only organized sports. Thus, it appears that the benefits associated with participation in both activities are similar to those related to participation in only organized sports, and that the benefits associated with self-organized PA are only present when young people concurrently engage in organized sports. 

### 4.4. Limitations

Regarding limitations, first, the cross-sectional nature of the present study prevents any causal relationship claim on the different associations that were studied. For instance, while “healthy” BMI can lead to increased participation in organized sports, it is also possible that increased participation in PA can lead to healthy BMI as Cairney and Veldhuizen [57] found in a longitudinal study. Second, there is the question of whether the general items that were used to measure the two activity types adequately captured and sufficiently differentiated the two activity types. Some activities that were reported as self-organized behaviours may probably be classified as organized behaviours (e.g., at fitness centres) with routines and structures similar to those of organized sports, something that differs from a self-organized play in a skateboard bowl [34]. These are measurement details that future studies can look into. Third, participants self-reported on items that measured their involvement in sports and PA, the developmental factors and demographics (some of which measured parental factors). While the reliability of such a subjective report can be questioned, it can be strengthened in future research by using objective measures (e.g., for engagement in the activity types) and supplementary responses from parents on items measuring parental factors. Fourth, BMI is not always seen as a good indicator of healthy weight [64]. Additional health measures that will effectively assess healthy growth in youth may be necessary to include in future studies. Fifth, several relevant factors (e.g., motivation and enjoyment) that can increase participation in sports and PA as well as reduce drop-out rates were not considered in the present study. We recommend that these factors be considered in future research.

### 4.5. Implications for Research, Policy and Practice

Our findings have implications for research, policy and practice. For research, the decrease in PA with age among young people is certainly not the only challenge to be tackled. While more research is needed to investigate this downward trend and to find out ways to curb it, there is also the need to look into the gender, socio-economic and majority/minority differences in sports and PA participation. Finding the relevant factors that will promote and maintain beneficial levels of PA in different youth groups can be one way of contributing to positive youth development. Besides, knowledge on how communities and nations can optimize adolescents’ developmental processes and thus promote social justice through PA should be seen as the goal of future PA research [34]. This type of research has to apply new theoretical approaches (e.g., Positive Youth Development [PYD] Theory) that are built on faith in youth and self-governing practices and that differs from theories on performance enhancement knowledge derived from epidemiology.

Furthermore, policies that will ensure that different kinds of activities are offered to meet the PA needs of young people across age, sex and socio-economic status are needed. At the school and community level, having a policy that involves young people from these subpopulations in the planning and implementation of the PA programmes would likely make the activities more effective. 

There are also some practical implications of our findings. We found that while organized sport was more likely to be associated with developmental factors, the protective role of self-organized PA appear to be reinforced by the organized sports activities that young people were concurrently involved in. In Norway, organized sports are usually arranged outside school hours. However, schools can still team up with the local communities and together use their facilities and services to offer structured and supervised PA programmes that will allow young people to reach the recommended PA goal as well as reap the associated health, social and cognitive benefits. Yet, we know from earlier research that youth drop out from organized sport and that not more than 50% of 15-year-olds engage in organized youth sports. According to Säfvenbom and colleagues [41], there are many reasons for this dropout, still also great possibilities to see more adolescents involved in movement activities outside competitive youth sports. The most widespread reason for involvement in movement activities is the fun and joy associated with the activity and for some adolescents, achievement, performance and competition do not represent the fun in sports. Thus, alternative channels for communicating alternative movement activities have to be developed also in ways that secure equal opportunities for all. One way to go about it will be to promote sports and PA through PYD interventions [65].

## 5. Conclusions

Both organized sports and self-organized PA were associated with several developmental benefits although the benefits of the former appeared to outweigh that of the latter. In addition, participation in both activity types appeared to be more developmentally beneficial than participation in only one activity, in particular, self-organized PA. While there are still measurement issues of the PA items and other limitations to be dealt with, females and older youth, and to some extent minorities and youth from low-income families, appeared to be less involved in PA. Organized sports are coordinated by the local community and outside school time programmes in Norway, yet schools can team up with other stakeholders to secure a PA programme that may help young people increase their PA levels. An efficient programme will be one that involves both organized sports and self-organized activities. Programmes where young people are also involved in the planning and implementation of the activities may be a step in the right direction.

## Figures and Tables

**Table 1 ijerph-16-00585-t001:** Frequency distribution of number of hours spent in organized sports and self-organized physical activity (PA).

Organized Sports	%	*n*
0 h	32.4	569
1–2 h per week	12.2	215
3–4 h per week	19.0	334
5–7 h per week	18.0	316
8–10 h per week	9.2	162
11 h or more per week	9.2	162
Total		1758
Frequency missing = 302		
Self-organized PA		
0 h	21.9	367
1–2 h per week	27.6	461
3–4 h per week	25.1	419
5–7 h per week	13.3	222
8-10 h per week	6.3	105
11 h or more per week	5.9	98
Total		1672
Frequency missing = 388		
Physical activity types		
None	11.3	182
Only organized PA for at least 1 h	10.8	174
Only self-organized PA for at least 1 h	20.3	326
Both for at least 1 h each	57.5	924
Total		1606
Frequency missing = 454		

**Table 2 ijerph-16-00585-t002:** Descriptive analysis and correlation analyses of study variables with organized sports and self-organized physical activity.

Study Variables	Correlation Coefficient																	
	2	3	4	5	6	7	8	9	10	11	12	13	14	15	16	17	18	19
1. Organized (Org) sports ^a^	0.30 **	0.06 *	0.75 **	−0.06 **	−0.17 **	0.10 **	−0.07 **	−0.08 **	0.35 **	−0.10 **	0.10 **	0.10 **	0.01	0.36 **	0.07 **	0.19 **	−0.18 **	−0.07 *
2. Self-organized (S-org) PA		0.58 **	−0.01	−0.12 **	−0.05	0.11 **	−0.06 *	−0.03	0.25 **	−0.01	0.09 **	0.06 *	0.03	0.17 **	0.04	0.14 **	−0.08 *	0.03
3. Both activities vs Org sports			^f^	0.00	−0.08 **	0.08 *	−0.02	−0.03	0.18 **	0.01	0.08 **	0.06	0.02	0.13 **	0.05	0.09 **	−0.09 **	−0.08 *
4. Both activities vs S-org PA				−0.02	−0.21 **	−0.01	0.01	−0.03	0.23 **	−0.09 **	0.03	0.01	−0.07 *	0.26 **	−0.03	0.11 **	−0.14 **	−0.13 **
5. Gender ^b^					−0.00	−0.06 *	−0.02	−0.01	0.01	−0.11 **	−0.03	0.00	0.08 **	−0.14 **	0.21 **	−0.00	0.11 **	0.06 *
6. Age						0.06 **	0.03	0.07 **	−0.32 **	0.04	−0.09 **	−0.09 **	0.07 **	0.04	0.07 **	−0.03	0.26 **	0.54 **
7. Parents’ total income ^c^							−0.13 **	−0.17 **	0.15 **	−0.01	0.17 **	0.20 **	0.19 **	0.13 **	0.18 **	0.14 **	−0.01	0.13 **
8. Mother’s ethnicity ^d^								0.63 **	−0.10 **	−0.01	−0.00	−0.05 *	−0.02	−0.08 **	−0.06 *	−0.05	−0.01	−0.12 **
9. Father’s ethnicity ^d^									−0.14 **	−0.01	−0.02	−0.07 **	−0.02	−0.07 *	−0.06 **	−0.04	−0.01	−0.09 **
10. Parental activity support										−0.06 *	0.25 **	0.14 **	−0.01	0.23 **	0.08 **	0.21 **	−0.27 **	−0.24 **
11. BMI ^e^											−0.06 *	−0.07 **	−0.05 *	−0.09 **	−0.09 **	−0.08 **	0.05	0.06 *
12. Academic competence												0.47 **	0.38 **	0.22 **	0.40 **	0.35 **	−0.25 **	−0.14 **
13. Mathematics grade													0.42 **	0.29 **	0.45 **	0.05 *	−0.25 **	−0.12**
14. English grade														0.21 **	0.55 **	0.08 **	−0.11 **	0.04
15. Physical education grade															0.28 **	0.20 **	−0.18 **	0.03
16. Norwegian language grade																0.11 **	−0.12 **	0.02
17. Social competence																	0.00	0.11 **
18. Smoking status																		0.49 **
19. Alcohol misuse																		-
*Descriptive analysis*																		
*Range*	1–6	0–1	0–1	1–2	13–18	1–5	1–2	1–2	1–4	1–2	1–4	1–6	1–6	1–6	1–6	1–4	1–4	1–3
*Mean*	2.72	0.78	0.71	1.52	15.29	2.99	1.11	1.12	2.14	1.15	2.84	3.81	4.14	4.57	4.22	3.16	1.58	1.75
*S.D.*	1.41	0.42	0.46	0.50	1.51	1.01	0.32	0.32	0.64	0.35	0.58	1.09	0.99	0.84	0.85	0.56	0.89	0.87

**Note:**^a^ Organized sports—(range: 1–6; mean (standard deviation (SD)): 2.87 (1.67)); ^b^ Gender—(1) Males and (2) Females; ^c^ (1) less than 200,000, (2) 200,000–400,000, (3) 400,000–800,000, (4) 800,000–1 million, (5) more than 1 million; ^d^ (1) Mother (father) is Norwegian and (2) Mother (father) is not Norwegian; ^e^ Cole et al. (2000) age- and gender-adjusted body mass index (BMI): (0) Not overweight or obese, (1) overweight or obese; ^f^ Was not computed due to the constant value in at least one variable; * *p* < 0.05; ** *p* < 0.01.

**Table 3 ijerph-16-00585-t003:** Ordinal regression analyses of organized and self-organized physical activity.

	Organized Sports				Self-Organized PA			
Study variables	Est.^d^	S.E.	Est./S.E.	Sig.	Est.^d^	S.E.	Est./S.E.	Sig.
Demographics Sex ^a^	0.01	0.02	0.40	0.689	−0.08	0.03	−3.29	0.001
Age	−0.21	0.03	−7.56	0.000	−0.10	0.03	−3.41	0.001
Parents’ total income	0.07	0.03	2.66	0.008	0.08	0.03	2.88	0.004
Mother’s ethnicity ^b^	0.00	0.03	0.05	0.963	−0.05	0.03	−1.52	0.129
Father’s ethnicity ^b^	−0.03	0.03	−1.10	0.274	0.01	0.03	0.46	0.649
Parental activity support	−0.00	0.02	−1.16	0.870	−0.01	0.02	−0.55	0.581
Developmental factors BMI ^c^	−0.05	0.03	−2.12	0.034	0.02	0.03	0.63	0.528
Academic competence	0.02	0.03	0.64	0.524	0.04	0.03	1.43	0.153
Mathematics grade	−0.05	0.03	−1.75	0.081	−0.03	0.03	−0.87	0.382
English grade	−0.06	0.03	−2.01	0.044	−0.03	0.03	−0.80	0.423
Physical education grade	0.38	0.03	13.92	0.000	0.15	0.03	4.87	0.000
Norwegian language grade	0.01	0.03	0.17	0.867	0.03	0.04	0.82	0.413
Social competence	−0.01	0.02	−0.26	0.794	0.00	0.04	0.06	0.949
Smoking status	−0.12	0.03	−4.25	0.000	−0.08	0.03	−2.41	0.016
Alcohol misuse	0.08	0.03	2.50	0.013	.10	.04	2.69	0.007

**Note:**^a^ Sex—(1) Male and (2) Female; ^b^ (1) Mother (father) is Norwegian and (2) Mother (father) is not Norwegian; ^c^ Cole et al. (2000) age- and sex-adjusted BMI: (0) Not overweight or obese, (1) overweight or obese; ^d^ Standardized coefficient.

**Table 4 ijerph-16-00585-t004:** Multiple logistic regression analyses: both organized sports and self-organized PA versus organized sports only or self-organized PA only.

	Organized Sports only vs. Both Activities					Self-Organized PA only vs. Both Activities				
Study variables	B	S.E.	Sig	OR	95% CI	B	S.E.	Sig	OR	95% CI
Demographics Sex ^a^	0.16	0.15	0.302	1.17	0.91–1.51	0.27	0.14	0.061	1.31	1.03–1.66
Age	−0.14	0.06	0.024	0.87	0.79–0.96	−0.38	0.06	0.000	0.69	0.63–0.75
Parents’ total income	0.19	0.09	0.031	1.20	1.05–1.38	−0.01	0.08	0.951	1.00	0.87–1.13
Mother’s ethnicity ^b^	0.10	0.27	0.705	1.11	0.71–1.71	0.35	0.27	0.189	1.42	0.92–2.20
Father’s ethnicity ^b^	−0.04	0.27	0.888	0.96	0.62–1.49	−0.23	0.25	0.347	0.79	0.53–1.19
Parental support	−0.00	0.00	0.546	1.00	1.00–1.00	0.00	0.00	0.750	1.00	1.00–1.00
Developmental factors BMI ^c^	0.20	0.24	0.409	1.22	0.82–1.81	−0.31	0.20	0.117	0.73	0.53–1.02
Academic competence	0.15	0.13	0.241	1.16	0.94–1.44	−0.01	0.12	0.919	0.99	0.81–1.20
Mathematics grade	−0.06	0.08	0.445	0.94	0.82–1.08	−0.15	0.08	0.060	0.86	0.76–0.98
English grade	−0.09	0.10	0.331	0.91	0.78–1.07	−0.21	0.09	0.022	0.81	0.70–0.94
Physical education grade	0.38	0.11	0.001	1.47	1.22–1.77	0.99	0.11	0.000	2.71	2.24–3.27
Norwegian language grade	0.06	0.12	0.625	1.06	0.87–1.29	−0.10	0.11	0.369	0.90	0.75–1.09
Social competence	0.00	0.00	0.355	1.00	1.00–1.00	0.00	0.00	0.775	1.00	1.00–1.00
Smoking status	−0.17	0.11	0.135	0.85	0.71–1.02	−0.30	0.10	0.002	0.74	0.63–0.87
Alcohol misuse	−0.05	0.13	0.726	0.95	0.77–1.19	0.12	0.10	0.260	1.12	0.95–1.34

**Note:**^a^ Sex—(1) Male and (2) Female; ^b^ (1) Mother (father) is Norwegian and (2) Mother (father) is not Norwegian; ^c^ Cole et al. (2000) age and sex adjusted BMI: (0) underweight or normal, (1) overweight or obese.

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
