# Peer review of "Participation in Organized Sports and Self-Organized Physical Activity: Associations with Developmental Factors"

_ijerph, 2019, doi:10.3390/ijerph16040585_

Round 1
Reviewer 1 Report
The manuscript entitled “Participation in organized sports and self-organized physical activity: Associations with developmental factors” is really very well-written and structured, providing an important contribution to previous literature and being in line with the topic of this special issue. Congratulations to the authors. Some remarkable strengths are described and some minor limitations might be underlined, in order to perform a very minor revision:
-Strenghts:
a) Big and representative sample
b) Very complete literature review and study justification
c) Detailed methods’ description.
d) Adequate data analysis, according to study aims
e) Clear and organized results’ presentation
f) Very thorough discussion, by presenting main contributions, hypotheses contrast, limitations, and implications for research, policy and practice.
-Minor suggestions to revise:
a) Hypotheses may be justified by including citations of the previous literature described in the introduction.
b) Little test may be conducted to check that missing values are randomly distributed.
c) Goodness of fit statistics should be added to describe regression analyses.
d) General study aims may introduce the discussion section.
e) Please, include citations to justify the limitations acknowledged.
f) As practical implications, interventions to promote PYD through sport may be cited.
Author Response
Many thanks for your comments. Please find below our responses to your comments which have been incorporated in our revision.
-Minor suggestions to revise:
a) Hypotheses may be justified by including citations of the previous literature described in the introduction.
Response: We have inserted citations to justify the hypothesis that was made. See page 3, line 144.
b) Little test may be conducted to check that missing values are randomly distributed.
Response: We did a Little’s MCAR test to examine if missing values were randomly distributed. P-value of less than .05 showed that missing values were completely missing at random.
c) Goodness of fit statistics should be added to describe regression analyses.
Response: Due to our ordered categorical outcome variable and the missing cases in our study variables, we needed to use the so-called “Montecarlo” integration in our analysis in Mplus. With these analytical choices, chi-square and related fit statistics are not available.
d) General study aims may introduce the discussion section.
Response: We have introduced the discussion section with the general aims of the present study.
e) Please, include citations to justify the limitations acknowledged.
Response: Thank you for the suggestion. We have added citations to justify the acknowledged limitations. See limitation section.
f) As practical implications, interventions to promote PYD through sport may be cited.
Response: We have incorporated this suggestion in the implication section.
Reviewer 2 Report
This study was conducted under the theme of "Participation in organized sports and self-organized physical activity: Associations with developmental factors".
Introduction
I have decided that the authors are appropriate for the research questions I have raised in the introduction. It is an interesting topic to analyze the relationship between physical activity and social and academic growth.
Research method
Organized sports and self-organized PA are measured separately. There is a question as to how a reasonable measure can be obtained by the self-response method. Please add a description of the reliability of the data collection results.
Result
The number of missing data is written in the result table, and the difference in the number of cases for each area is easily understood. The non-parametric method of multiple regression analysis was applied to match the characteristics of the data. However, it would be nice to edit it so that there are no spaces on page 9.
Discussion
They divide the areas and describe the limitations and problems of this study with reasonable content.
Author Response
Thank you for your comments. Please find below our responses to your comments which have been incorporated in our revision.
a) Introduction
I have decided that the authors are appropriate for the research questions I have raised in the introduction. It is an interesting topic to analyze the relationship between physical activity and social and academic growth.
Response: Thank you for your comment!
b) Research method
Organized sports and self-organized PA are measured separately. There is a question as to how a reasonable measure can be obtained by the self-response method. Please add a description of the reliability of the data collection results.
Response: We have discussed this self-response method as a limitation that can be addressed in future studies. Please see limitation section.
c) Result
The number of missing data is written in the result table, and the difference in the number of cases for each area is easily understood. The non-parametric method of multiple regression analysis was applied to match the characteristics of the data. However, it would be nice to edit it so that there are no spaces on page 9.
Response: I think the spaces are technical errors that will be corrected later by the journal.
d) Discussion
They divide the areas and describe the limitations and problems of this study with reasonable content.
Response: Thank you for the comment.
Reviewer 3 Report
The manuscript entitled “Participation in organized sports and self-organized physical activity: Associations with developmental factors” analyses the influence of demographic and developmental factors to contribute to the current discussion about physical activity levels of adolescents. This study adds interesting new insights in what influences adolescents’ participation in various forms of physical activity and sports and should be shared with the research community and practitioners.
However, I suggest some revisions to improve the comprehensibility of the argumentation in this manuscript.
Firstly, the research gap should be highlighted in a clearer way. I would recommend to end every section in the introduction with a short summary about what we already know from current literature and which information is still missing. This helps to understand the argumentation for the research gap. For example in section “1.1. Involvement in Physical Activity: The Role of Demographic Factors” should summarize why gender (l. 53), ethnicity (l. 59) as well as social class and parental support (l. 69) might influence physical activity. This summary should clarify, why you have chosen these factors in your study.
Another example can be given in section “1.2. Physical Activity Types”. Here, I have had the impression that this section addresses different physical activity types until l. 93. From l. 94 following, information about influencing developmental aspects is given. I would recommend to introduce the section from l. 94 following with an extra subtitle to make clear that developmental factors are explained here. Additionally, a short summary after every developmental aspect might help to identify the research gap.
In section “1.2. The Present Study”, it might improve the transition from the research gap to your aim of the study, if the research question is given at the beginning of the paragraph in l. 135. The research question should pick up the research gap and makes it easier to understand in the following method section why you did what you did.
Secondly, the results section should be proofread carefully to avoid incongruences between tables and text. Throughout the results section, there are some different values when comparing the information given in the tables and in the text. For example:
- l. 249, r-values
- l. 257, r-values
- l. 261, r-values
Please check that the values given in the tables are the same in your text.
In the notes of the tables, gender is coded differently: In table 2 it is “male” or “female” whereas the following tables define gender as “boys” or “girls” Is there a special reason for this?
I am wondering where to find the footage “e” in table 3 (“Est.e”). Please check, if there is a footage missing in the notes of table 3.
Thirdly, in the–already well written–discussion section, the own results should be discussed a bit more in detail. I expected a discussion of the effect values displayed in the regression analyses when discussing the different demographic and developmental factors. As your results show, some factors have a significant influence on participation in physical activity or sports. It is good to know, which factors have an impact on participation in physical activities or sports but it is also very important to know which factors do have a higher or lower impact. This should be discussed with possible explanations in contrast with current literature to give hints for future research.
In some parts the discussion the writing is descriptive. Here, possible explanations in contrast to existing literature might be interesting for your discussion. For example in the section l. 355 to l. 363, I got curious about how you would explain the relation of parental support and adolescents’ physical activity: Why is there a decrease of parents’ support in older adolescents compared to younger ones? A relation between parents’ nationalities and the participation in organized sports is indicated but how could this be explained? Another example is the part where developmental factors are discussed (l. 364 to l. 376). It might be interesting for future research
to add some ‘hypothetical’ explanations to the presented associations.
At the end of the discussion section (l. 402), I expected an outlook on other possible factors that might explain adolescents’ participation in physical activities and sports. It might be worth to discuss personal factors, for example motivation or enjoyment in contrast to your findings (as you mentioned later in “4.5. Implications for Research, Policy and Practice”). I would recommend to weigh up developmental and demographic factors against personal factors concerning the question which factors should be emphasized to best reduce drop-out rates of adolescents.
Last but not least, here are some minor revisions that can be addressed in a final proofreading:
- l. 65 to 67: Is there a difference between children and adolescents?
- L. 71 to 76: Is this paragraph necessary to understand the argumentation later on?
- L. 135: Why do you choose gender? In the manuscript, there is no differentiation of gender but sex (male–female)
- L. 168: Adolescents reports: This might be a bias that should be discussed in the limitation section.
- L. 182, l. 196, l. 206 following: The Cronbach’s alpha values are acceptable but below .85. Is this a limitation for your study? If yes, please discuss. If not, please clarify.
The authors should be encouraged to revise this manuscript since it contributes interesting new associations between different factors that influence adolescents’ physical activity. I hope that these recommendations help the authors during the revision process and I am looking forward to the revised version of the manuscript.
Author Response
Many thanks for your insightful comments that have helped to improve the paper. Please find below our responses to your comments which have been incorporated in our revision.
1a) Firstly, the research gap should be highlighted in a clearer way. I would recommend to end every section in the introduction with a short summary about what we already know from current literature and which information is still missing. This helps to understand the argumentation for the research gap. For example in section “1.1. Involvement in Physical Activity: The Role of Demographic Factors” should summarize why gender (l. 53), ethnicity (l. 59) as well as social class and parental support (l. 69) might influence physical activity. This summary should clarify, why you have chosen these factors in your study.
Response: At the beginning of section “1.1. Involvement in Physical Activity: The Role of Demographic Factors”, we mention that gender, ethnicity, social class and parental support for physical activity have been found to play a relevant role in youth’s participation in physical activity in earlier research. Our goal is to determine whether this is the case for our Norwegian sample. We have added this information at the end of that section.
1b) Another example can be given in section “1.2. Physical Activity Types”. Here, I have had the impression that this section addresses different physical activity types until l. 93. From l. 94 following, information about influencing developmental aspects is given. I would recommend to introduce the section from l. 94 following with an extra subtitle to make clear that developmental factors are explained here. Additionally, a short summary after every developmental aspect might help to identify the research gap.
Response: In our revision, we have labelled section 1.2. as “Physical Activity Types and Developmental Factors” and have added a brief summary that shows the state of research at the end of the section.
1c) In section “1.2. The Present Study”, it might improve the transition from the research gap to your aim of the study, if the research question is given at the beginning of the paragraph in l. 135. The research question should pick up the research gap and makes it easier to understand in the following method section why you did what you did.
Response: Thank you for this suggestion. In our revision, we include information that will improve the transition from the research gap to our aim. Please see section 1.3.
2a) Secondly, the results section should be proofread carefully to avoid incongruences between tables and text. Throughout the results section, there are some different values when comparing the information given in the tables and in the text. For example:
- l. 249, r-values
- l. 257, r-values
- l. 261, r-values
Please check that the values given in the tables are the same in your text.
Response: We are sorry for these inconsistencies and have revised accordingly.
2b) In the notes of the tables, gender is coded differently: In table 2 it is “male” or “female” whereas the following tables define gender as “boys” or “girls” Is there a special reason for this?
Response: Thank you for this observation. We have corrected the inconsistencies in our revision.
2c) I am wondering where to find the footage “e” in table 3 (“Est.e”). Please check, if there is a footage missing in the notes of table 3.
Response: “e” should have been “d”. We have revised this now.
3a) Thirdly, in the–already well written–discussion section, the own results should be discussed a bit more in detail. I expected a discussion of the effect values displayed in the regression analyses when discussing the different demographic and developmental factors. As your results show, some factors have a significant influence on participation in physical activity or sports. It is good to know, which factors have an impact on participation in physical activities or sports but it is also very important to know which factors do have a higher or lower impact. This should be discussed with possible explanations in contrast with current literature to give hints for future research.
Response: We have tried to address the issue of the factors that appear to have higher or lower impact as well as given hints for future research. Please see discussion section.
3b) In some parts the discussion the writing is descriptive. Here, possible explanations in contrast to existing literature might be interesting for your discussion. For example in the section l. 355 to l. 363, I got curious about how you would explain the relation of parental support and adolescents’ physical activity: Why is there a decrease of parents’ support in older adolescents compared to younger ones? A relation between parents’ nationalities and the participation in organized sports is indicated but how could this be explained? Another example is the part where developmental factors are discussed (l. 364 to l. 376). It might be interesting for future research to add some ‘hypothetical’ explanations to the presented associations.
Response: In our revision, we have extended the discussion by including some possible explanations regarding the observed associations. See discussion section.
3c) At the end of the discussion section (l. 402), I expected an outlook on other possible factors that might explain adolescents’ participation in physical activities and sports. It might be worth to discuss personal factors, for example motivation or enjoyment in contrast to your findings (as you mentioned later in “4.5. Implications for Research, Policy and Practice”). I would recommend to weigh up developmental and demographic factors against personal factors concerning the question which factors should be emphasized to best reduce drop-out rates of adolescents.
Response: Thank you for this comment. We have added some information regarding these relevant factors in our revision. Please see the limitation section.
4) Last but not least, here are some minor revisions that can be addressed in a final proofreading:
- l. 65 to 67: Is there a difference between children and adolescents?
Response: We used children and adolescents for clarity sake although in principle they may not be different.
- L. 71 to 76: Is this paragraph necessary to understand the argumentation later on?
Response: To some extent, as we pick it up again in the implication section.
- L. 135: Why do you choose gender? In the manuscript, there is no differentiation of gender but sex (male–female)
Response: We have chosen to focus on sex now, as that is the concept we describe in the paper.
- L. 168: Adolescents reports: This might be a bias that should be discussed in the limitation section.
Response: We have addressed adolescents reports now in the limitation section.
- L. 182, l. 196, l. 206 following: The Cronbach’s alpha values are acceptable but below .85. Is this a limitation for your study? If yes, please discuss. If not, please clarify.
Response: In the literature, Cronbach’s alpha of .70 or higher is considered “acceptable” in social science research so we did not see it as a limitation to address.
Round 2
Reviewer 3 Report
The authors adressed the comments very well.
Thank you very much for your clarifications!